# The Mechanisms Underlying Autonomous Adrenocorticotropic Hormone Secretion in Cushing’s Disease

**DOI:** 10.3390/ijms21239132

**Published:** 2020-11-30

**Authors:** Hidenori Fukuoka, Hiroki Shichi, Masaaki Yamamoto, Yutaka Takahashi

**Affiliations:** 1Division of Diabetes and Endocrinology, Kobe University Hospital, 7-5-2, Kusunoki-cho, Chuo-ku, Kobe 650-0017, Japan; yamamasa@med.kobe-u.ac.jp; 2Division of Diabetes and Endocrinology, Kobe University Graduate School of Medicine, 7-5-2, Kusunoki-cho, Chuo-ku, Kobe 650-0017, Japan; shichihi98@gmail.com (H.S.); takahash@naramed-u.ac.jp (Y.T.); 3Department of Diabetes and Endocrinology, Nara Medical University, 840 Shijo-cho, Kashihara, Nara 634-8522, Japan

**Keywords:** adrenocorticotrophic hormone, Cushing’s disease, glucocorticoid resistance, ectopic ACTH syndrome

## Abstract

Cushing’s disease caused due to adrenocorticotropic hormone (ACTH)-secreting pituitary adenomas (ACTHomas) leads to hypercortisolemia, resulting in increased morbidity and mortality. Autonomous ACTH secretion is attributed to the impaired glucocorticoid negative feedback (glucocorticoid resistance) response. Interestingly, other conditions, such as ectopic ACTH syndrome (EAS) and non-neoplastic hypercortisolemia (NNH, also known as pseudo-Cushing’s syndrome) also exhibit glucocorticoid resistance. Therefore, to differentiate between these conditions, several dynamic tests, including those with desmopressin (DDAVP), corticotrophin-releasing hormone (CRH), and Dex/CRH have been developed. In normal pituitary corticotrophs, ACTH synthesis and secretion are regulated mainly by CRH and glucocorticoids, which are the ACTH secretion-stimulating and -suppressing factors, respectively. These factors regulate ACTH synthesis and secretion through genomic and non-genomic mechanisms. Conversely, glucocorticoid negative feedback is impaired in ACTHomas, which could be due to the overexpression of 11β-HSD2, HSP90, or TR4, or loss of expression of CABLES1 or nuclear BRG1 proteins. Genetic analysis has indicated the involvement of several genes in the etiology of ACTHomas, including *USP8*, *USP48*, *BRAF*, and *TP53*. However, the association between glucocorticoid resistance and these genes remains unclear. Here, we review the clinical aspects and molecular mechanisms of ACTHomas and compare them to those of other related conditions.

## 1. Introduction

Cushing’s disease is characterized by hypercortisolemia occurring due to autonomous secretion of adrenocorticotropic hormone (ACTH) from a pituitary tumor. However, the mechanisms underlying the impaired physiological hormonal regulation in ACTH-secreting pituitary corticotroph adenomas (ACTHomas) remain unclear. In response to diurnal variation and stress, the synthesis and secretion of ACTH in normal pituitary corticotroph cells is stimulated by the hypothalamic neuroendocrine hormones, namely corticotrophin-releasing hormone (CRH) and arginine vasopressin (AVP). Conversely, ACTH synthesis and secretion are suppressed mainly through the action of glucocorticoid derived from the adrenal cortex. This negative feedback regulation occurs not only in corticotrophs within the pituitary, but also in hypothalamic CRH-secreting neurons [1]. ACTHomas maintain high ACTH levels as they respond to CRH and AVP, which can be evaluated using the CRH and 1-deamino-8-d-arginine vasopressin (DDAVP) tests, respectively. The impaired ability of glucocorticoids to suppress ACTH secretion is clinically proven to occur through inappropriate ACTH secretion with non-suppressive cortisol secretion after dexamethasone (Dex) treatment, and this functional assay is named as the ‘Dex suppression test’ (DST). These provocative and inhibitory tests are used to screen the patients for Cushing’s syndrome and act as a diagnostic tool to differentiate it from ectopic ACTH syndrome (EAS) or non-neoplastic hypercortisolemia (NNH, also known as pseudo-Cushing’s syndrome) [2,3]. Although the clinical significance of these tests is well established, knowledge regarding the molecular mechanisms underlying the pathophysiological regulation of ACTH in these conditions is limited, which is partially attributed to the lack of availability of human ACTHoma cell lines. 

In this review, we mainly focus on the mechanisms underlying ACTH synthesis and secretion in the normal pituitary, ACTHomas, and other related conditions. 

## 2. Material and Methods

To search the bibliographic references, online databases, including PubMed, Web of Science, SCOPUS, and Google were used. The following MeSH terms and their combinations were used for searching the references: Cushing, Cushing’s disease, POMC, ACTH secretion, ACTH synthesis, USP8, pseudo Cushing syndrome, ectopic ACTH syndrome, glucocorticoid negative feedback, DDAVP, CRH, and stress.

## 3. Results

### 3.1. Physiological Regulation of ACTH

#### 3.1.1. ACTH Synthesis and Secretion

ACTH is derived mainly from the pituitary corticotroph cells, in which a 266-amino-acid precursor protein, pro-opiomelanocortin (POMC), is translated from the *POMC* gene located on chromosome 2q23. The 8 kb long human *POMC* gene comprises 3 exons. In corticotrophs, POMC is mostly encoded by exon 3 of the gene. The *POMC* transcriptional activity is thought to be regulated mainly via the orphan nuclear receptor, Nurr77, and also Tpit, Pitx, NeuroD1, signal transducer and activator of transcription 3 (STAT3), and ETS variant transcription factor 1 (Etv1) [4]. It is modulated by pituitary-specific enhancer, which is located at −7 kb from the *POMC* initiation site [5], and also epigenetic modification, such as chromatin remodeling [6]. POMC protein is thereafter sorted into the dense-core secretory granules of the regulated secretory pathway (RSP). The N-terminal of POMC (1–26 aa) possesses a sorting signal motif, which helps POMC in binding to the membrane-associated form of carboxypeptidase E (CPE), a sorting receptor of POMC, thereby leading to normal post-translational protein processing [7]. In CPE knockout mice (*Cpe^fat/fat^* mice), POMC processing is reduced in both the pituitary and hypothalamus [8]. Secretogranin III (SgIII) is a sorting receptor of chromogranin A, which also binds to POMC in the pituitary [9,10]. In SgIII-deficient corticotroph cells, POMC has been shown to be accumulated in the trans-Golgi network (TGN), suggesting its important role in POMC trafficking. In the regulation of trafficking of POMC to the RSP, two proteins, including CPE and SgIII, have been shown to play a synergistic role and compensate for each other [11]. POMC is enzymatically cleaved into N-POMC (1–77), ACTH (1–39), and β-lipotropin (β-LPH) (1–89). Post-translational POMC processing by prohormone convertases (PCs) is required to produce ACTH. Corticotrophs predominantly express PC1 and PC3, which cleave POMC to generate ACTH (1–39) (Figure 1), while PC2 (expressed in melanotrophs) further cleaves ACTH into α-melanocyte-stimulating hormone (MSH) [12,13,14]. POMC processing is also modulated by other enzymes, including Yapsin A, ACTH-converting enzyme (AACE), aminopeptidases B-like (AMB), and peptidylglycine α-amidating monooxygenase (PAM) [15]. POMC also expresses in non-pituitary tissues, including hypothalamus, testis, adrenal gland, pancreas, adipose tissue, kidney, and placenta [16,17,18,19,20,21]. Following its synthesis, ACTH is stored in cytoplasmic secretory vesicles. The secretion of ACTH is predominantly mediated through non-genomic mechanisms, including calcium- and voltage-activated potassium (BK) channels [22,23], and annexin 1 (ANXA1).

#### 3.1.2. Positive Regulation of ACTH Synthesis and Secretion

ACTH synthesis is mainly stimulated through the action of CRH, which is a trophic hormone synthesized in the hypothalamic paraventricular nucleus (PVN) and secreted into hypophyseal portal vessels at the median eminence. CRH binds to CRH receptor 1 (CRHR1), which is expressed on the cellular membrane of corticotrophs, further triggering the accumulation of cAMP, activation of protein kinase A (PKA), and subsequent *POMC* transcription [24,25]. Although the classical PKA pathway to Ca^2+^/cAMP response-element binding protein (CREB) is involved in *POMC* transcription [26], Nur77 (also known as NGFI-B) appears to play a dominant role [27,28,29]. Although Nur77-binding response element (NBRE) is present in the *POMC* promoter, another motif, Nur response element (NurRE), is also required for Nur77-dependent *POMC* mRNA production [30]. The homo- or hetero-dimerization of Nur77 with other Nur family members, such as Nurr1 or NOR1, is required for its translocation into the nucleus and binding to NurRE. This cascade is mainly induced through MAPK downstream of PKA [30]. PKA also induces cytosolic Ca^2+^ accumulation through calcium-dependent voltage channels, thereby resulting in calmodulin kinase II (CaMKII) activation. This Ca^2+^-dependent pathway stimulates distinct downstream signaling, including Nurr induction and MAPK-mediated activation of Nurr [31]. Conversely, during CRH-mediated *POMC* expression, a pituitary-specific enhancer located 7 kb upstream of the initiation site has been shown to play a pivotal role [5]. In regard to ACTH secretion, CRH rapidly stimulates the burst of Ca^2+^ cytosolic inclusion with large-conductance BK channels in a concentration-dependent manner [22,23,32].

AVP, a cyclic nonapeptide, is a main regulator of water homeostasis via the receptor AVPR2, located in the renal tubules. Furthermore, AVP triggers vasoconstriction via AVPR1a, which is predominantly expressed on vascular smooth muscle cells [33]. Additionally, AVP is known to be an important modulator of the HPA axis, especially during stress. AVP is synthesized in the hypothalamic PVN and supraoptic nuclei (SON) and stored in the posterior pituitary lobe. It is released into the hypophysial portal vein and can reach the anterior lobe of the pituitary [34], where it binds to AVPR1b on the pituitary corticotrophs to induce the ACTH secretion process. Intracellular signaling stimulates a biphasic release of cytosolic Ca^2+^ from intracellular inositol 1,4,5-triphosphate (IP_3_)-sensitive stores, and extracellular Ca^2+^ via L-type Ca^2+^ channels [25,35,36]. CRH, but not AVP, plays a major role in ACTH secretion under non-stress conditions. In fact, basal ACTH levels are similar in both *AVP*- and *AVPR1b*-deficient Brattleboro rats when compared to wild type [37,38]. Conversely, *AVPR1b* knockout mice exhibit dampened ACTH response to acute stress. Moreover, AVP storage and expression of AVPR1b receptor gradually increase during chronic stress, suggesting that AVP plays a primary role in adaptation to stress through enhancing the responses at the HPA axis [38,39,40]. Furthermore, ACTH secretion induced by AVP is mediated via BK-independent pathways [22].

Several cytokines, such as interleukin-1β (IL-1β), IL-6, and leukemia inhibitory factor (LIF) can stimulate ACTH secretion via direct or indirect mechanisms [41]. LIF and IL-6 are derived from hypothalamus and pituitary in response to inflammation, and act in an autocrine and paracrine manner on corticotrophs, respectively [42]. These proinflammatory cytokines induce STAT3 activation upon translocating to the nucleus, which is followed by binding to a STAT binding element located at position –399/–379 in the rat *Pomc* promoter. STAT3 activated through LIF or IL-6 stimulates co-expression of *POMC* with CRH in a synergistic manner [43,44]. In response to inflammation, microenvironmental protons increase and induce *POMC* expression through the CRHR1/CaMKII pathway. Interestingly, this induction occurs without ligand stimulation of CRHR1 [45]. Urocortin is a CRH-related peptide that binds to both type 1 and type 2 CRH receptors, leading to the synthesis and secretion of ACTH via the cAMP and protein kinase C (PKC) pathway [46,47]. 

#### 3.1.3. Negative Regulation of ACTH Synthesis and Secretion

Glucocorticoid is a major factor that suppresses ACTH synthesis and secretion through its action on both the hypothalamic CRH neuron and pituitary corticotrophs. This is the predominant negative feedback system for the functioning of ACTH within the HPA axis [48], with both rapid non-genomic and delayed genomic mechanisms. In hypothalamic CRH neurons, both membrane-associated G protein-coupled glucocorticoid receptor (GR_mb_) and nuclear glucocorticoid receptor (GR) are expressed abundantly. As a non-genomic effect, glucocorticoid binds to GR_mb_, thereby leading to the release of endocannabinoid (CB) from neurons. CB binds to the cannabinoid receptor (CBR) located on the presynaptic glutamate terminal, thereby suppressing glutamine release in the PVN neuron, resulting in decreased CRH secretion [49]. GR binds directly to the negative GR element (nGRE) in DNA [50], and it also suppresses forskolin-dependent CRH activity via CRE in the *CRH* promoter [51]. The genomic effect of glucocorticoid in CRH neurons is still debatable. In corticotrophs, glucocorticoids rapidly suppress ACTH secretion by reducing both spontaneous and CRH-induced burst firing in a BK channel-mediated manner [52]. ACTH secretion from corticotrophs is also suppressed by Annexin 1 (ANXA1), secreted from the pituitary folliculostellate cells. In fact, glucocorticoids induce ANXA1 expression and its translocation to the outer surface of the plasma membrane of corticotrophs, suggesting that ANXA1 mediates glucocorticoid-dependent suppression of ACTH secretion [53]. As part of the genomic mechanism of glucocorticoid-dependent *POMC* suppression, there are several sites in the *POMC* promoter that might contribute to the glucocorticoid-dependent transcriptional suppression of ACTH. These sites include a negative glucocorticoid response element (nGRE), a direct GR binding site, an E-box, and a neurogenic differentiation factor 1 (NeuroD1) binding site [54,55]. However, glucocorticoid-mediated ACTH suppression is not mediated through direct binding of GR to the *POMC* promoter, but through its interaction with Nur factors, thereby inducing protein-protein interaction-dependent transrepression [56,57]. This process requires Brg1, the ATPase subunit of the Swi/Snf complex, which stabilizes the interaction between GR, Nur77, and HDAC2 on the *POMC* promoter [6] (Figure 2). The TGF-β superfamily member bone morphogenetic protein (BMP)4 has been implicated in pituitary organogenesis and differentiation. BMP4 suppresses *POMC* expression through preventing the binding of Tpit and Pitx1 transcription factors to the *POMC* promoter [58,59,60]. Furthermore, through its binding to somatostatin receptor subtype 2 (SSTR2) and subtype 5 (SSTR5), hypothalamic somatostatin (SST) suppresses the secretion of several pituitary hormones, including GH, TSH, and ACTH [61,62,63,64]. SST also suppresses intracellular cAMP levels and inhibits exocytosis through reducing cytosolic Ca^2+^ inclusion [65,66]. Constitutively activated SSTR5 attenuates both CRH-dependent increase in intracellular cAMP levels and ACTH secretion via posttranscriptional suppression of CRHR1 [67]. 

### 3.2. Pathological Regulation of ACTH in ACTHomas

#### 3.2.1. ACTHomas Tumorigenesis 

Pituitary adenomas, such as ACTHomas, are of monoclonal origin, suggesting that a single somatic gene abnormality might lead to their incidence [68,69,70]. Until recently, there were only few reports regarding the genetic changes in ACTHomas, except for familial Cushing’s disease, such as multiple endocrine neoplasm type 1 (MEN1), MEN4, and familial isolated pituitary adenomas (FIPAs), and the germline mutations associated with these hereditary syndromes were identified in the *MEN1*, *CDKN1B*, and *AIP* genes, respectively. However, these hereditary syndromes account for less than 3% of pituitary adenomas and do not have specific association with ACTHomas [71,72,73]. A major milestone in the understanding of ACTHoma pathology was achieved with the identification of somatic mutations in the *ubiquitin-specific-protease 8* (*USP8*) gene. Mutations in this gene have been found in 20–60% of ACTHomas cases [74,75]. All point mutations are specifically located within the 14-3-3 binding motif of USP8, and lead to an increase in the deubiquitylation activity of this enzyme. The epidermal growth factor receptor (EGFR) tyrosine kinase, which is frequently overexpressed in ACTHomas, is one of the deubiquitination substrates of USP8 [76]. Moreover, corticotroph-specific EGFR overexpression leads to ACTHomas mediated via E2F1 activation [77,78]. Recently, a de novo germline *USP8* mutation (c.2155T>C, p.S719P) in the 14-3-3 binding motif has also been reported in pediatric patients with Cushing’s disease [79]. Furthermore, subsequent analysis using next-generation sequencing led to the identification of somatic mutations in *USP48*, *BRAF*, and *TP53* genes, which were considered to be the other causes of this disease [80,81,82]. *USP48* activates NF-κB, which in turn binds to and transactivates the *POMC* promoter [80]. Almost 13% of ACTHomas cases have been shown to harbor specific p.Met415 mutations within the catalytic domain of USP48 [82], indicating a functional relevance of this ubiquitin-specific protease. The *BRAF* p.V600E mutation has been found in several cancers, including melanoma, papillary thyroid carcinomas, colon cancer, gliomas, and papillary craniopharyngiomas [83,84,85,86,87]. This activating mutation causes increased cell proliferation due to the activation of the MAPK pathway. In Cushing’s disease, MAPK activation can also induce ACTH synthesis, which leads to autonomous ACTH secretion. This mutation accounts for approximately 7% of cases with Cushing’s disease [82]. *TP53* is a well-recognized tumor suppressor gene, which guards the genome and regulates apoptosis, DNA repair, and cellular senescence. Loss-of-function mutations in the *TP53* gene are present in approximately 12.5% of ACTHomas cases and are associated with aggressive tumor behavior. Another gene associated with aggressive ACTHomas is *CABLES1*, a negative regulator of the cell cycle that interacts with cyclin-dependent kinase 3. *Cables1* has been identified as a glucocorticoid-dependent cell cycle modulator in AtT20, a cell line derived from a mouse with ACTHomas. Low or undetectable expression level of *CABLES1* is a distinct feature of ACTHomas in pediatric or young adult patients with Cushing’s disease, whereas the gene expresses abundantly in normal corticotrophs [88].

In pediatric patients with Cushing’s disease, mutations in *DICER1*, a small RNA processing endoribonuclease that cleaves precursor microRNAs (miRNAs) into mature miRNAs, lead to DICER1 syndrome [89]. This syndrome is associated with the development of pleuropulmonary blastomas, cystic nephroma, rhabdomyosarcoma, and several endocrine neoplasms, including thyroid cancer, ovarian tumors, and ACTH-secreting pituitary blastoma [90,91], thereby suggesting a pathogenic role for microRNAs in ACTHomas. Additionally, whole exome sequencing (WES) analysis revealed several other mutations associated with ACTHomas, including those in the *NR3C1*, *DAXX*, *ATRX*, and *HCFC1* genes [82]. Somatic mutations in the *MEN1*, *PRKAR1A*, and *GNAS1* genes have also been reported in ACTHomas [92,93]. Although several genes that are associated with cAMP signaling or the cell cycle have therefore been implicated in the pathogenesis of ACTHomas, the precise molecular mechanisms through which these mutations lead to tumorigenesis remain unclear. 

#### 3.2.2. ACTH Regulation in ACTHomas

Based on the response to low-dose DST, one of their most prominent characteristics of ACTHomas is the impaired ACTH suppression by glucocorticoid negative feedback [94], which further leads to ACTH-dependent hypercortisolemia. The decreased sensitivity to glucocorticoids cannot be explained by the loss-of-function mutations in *GR* in most cases [95]. At the intracellular pre-receptor level, glucocorticoid activity has been shown to be negatively regulated through 11β-hydroxysteroid dehydrogenase type 2 (11β-HSD2), which converts cortisol into its inactive form, cortisone. Increased expression of 11β-HSD2 has been observed in ACTHomas, and is in turn implicated in inducing glucocorticoid resistance [96,97]. Another mechanism possibly underlying glucocorticoid resistance is the increased expression of the chaperone protein, heat shock protein 90 (HSP90) [98,99]. GR is mainly cytoplasmic in its inactive state associated with HSP90, which regulates its intracellular trafficking, folding, maturation, and activation [100]. Silibinin, an inhibitor that binds to the C-terminal domain of HSP90, enhanced glucocorticoid sensitivity and suppressed ACTH secretion, suggesting that the increased level of HSP90 in ACTHomas plays an important role in impaired glucocorticoid negative feedback [98]. Additionally, a SWI/SNF protein, Brg1 and histone deacetylase 2 (HDAC2). are associated with the *POMC* promoter. Brg1 stabilizes the interaction between GR and HDAC2 to suppress *POMC* transcription. Loss of either nuclear Brg1 or HDAC2 can lead to glucocorticoid resistance in ACTHoma [101]. Testicular receptor 4 (TR4), an orphan nuclear receptor, interacts directly with the N-terminal domain of GR, resulting in the disruption of GR binding to the *POMC* promoter [102]. Therefore, an increase in the expression of TR4 is thought to contribute to glucocorticoid resistance in ACTHomas [103]. Using the murine AtT20 cell line, *Cables1* was identified as a glucocorticoid-responsive cell cycle regulatory gene, as described above in the tumorigenesis of ACTHomas. In 55% of ACTHomas cases, loss of expression of *CABLES1* has been reported, resulting in impaired sensitivity to glucocorticoids [104]. Conclusively, increased expression of 11β-HSD2, HSP90, or TR4, and loss of expression of BRG1 or CABLES1, contribute to the pathogenesis of ACTHomas. This mainly occurs through conceding GR function, which in turn impairs the glucocorticoid negative feedback system. However, the precise mechanism through which the expression of these molecules is altered, and the link between these molecules and other genetic changes remains unclear. In addition to the impaired glucocorticoid negative feedback, ACTHomas exhibits increased or aberrant expression of AVPR1b or AVPR2, which is associated with a paradoxical response to DDAVP [105,106]. Elevated expression levels of EGFR have been associated with high ACTH synthesis and secretion via E2F1-mediated transcriptional activity, which has been shown to be attenuated through the application of its tyrosine kinase inhibitor, gefitinib, and E2F1 inhibitor, HLM006474 [77,78]. The newly developed high-throughput “ACTH AlphaLISA assay” led to the identification a dual PI3K/HDAC inhibitor, CUDC-907, as a potential targeted therapy for ACTHomas. It was thought to be mediated via Nurr1 transcriptional activity in corticotroph adenomas, suggesting an important role of HDAC action in addition to PI3K in ACTHomas [107]. Furthermore, epigenetic-targeting compound, JQ1, which is an inhibiter of bromo and extra-terminal domain (BET), and targets bromodomain of the protein family members BRD2, BRD3, BRD4, and BRDT, has been identified to suppress POMC expression, supporting the important role of epigenetic control in ACTH synthesis in ACTHomas [108].

### 3.3. Pathological Regulation of Acth under Other Conditions

#### 3.3.1. Ectopic ACTH Syndrome

Ectopic ACTH syndrome (EAS) is a rare form of ACTH-dependent Cushing’s syndrome that occurs at a frequency of approximately 12% [109,110]. Extra-pituitary ACTH hypersecretion commonly occurs in neuroendocrine tumors of various tissue types, including small-cell lung carcinomas (SCLCs), bronchial carcinoids, thymic neuroendocrine neoplasms (NENs), pheochromocytomas, and medullary thyroid carcinomas [111,112,113,114,115,116,117,118,119,120,121]. Although the mechanism of EAS tumorigenesis is not fully understood, several genetic abnormalities have been identified in thymic NEN, including *HRAS*, *PAK1*, and *MEN1* using whole-exome sequencing [122]. The pathogenesis of NENs has been widely investigated, including the ones originating from lungs, gastrointestinal, and pancreatic tissues, showing genetic abnormalities. These involve DNA damage repair, chromatin remodeling, mTOR signaling, and telomere maintenance-related genes. In addition to the genetic alterations, epigenetic modification has been described, including DNA methylation and histone modifications [123,124]. However, these might be associated only with tumorigenesis but not with the incidence of autonomous hormone secretion. Regarding the molecular pathology of ectopic *POMC* expression in NENs, ACTH synthesis is not regulated through the action of pituitary-specific transcriptional factors, namely Pitx1 and Tpit; however, it might be enhanced by the phosphorylation-dependent DNA binding of E2F1 at the proximal region of the *POMC* promoter [125]. The detailed mechanisms underlying the ectopic expression of ACTH in EAS remain to be elucidated. However, in case of ACTH-secreting pheochromocytoma, a paradoxical increase in ACTH levels post-glucocorticoid administration was observed, which was found to be mediated by demethylation of the E2F-binding site in the *POMC* promoter [126]. In these tumors, immature ACTH precursors are frequently released in the circulation, which may contribute to the relatively high concentration of ACTH levels (100–200 pM/454.1–908.2 pg/mL), while the concentration in patients with Cushing’s disease is generally found to be less than 100 pM (454.1 pg/mL) [127,128,129]. In these NENs, the expression of POMC-processing enzymes, PC1 and PC3, is limited, probably due to impaired differentiation. For some carcinoids, ectopic PC2 expression can be explained as a reason for the secretion of corticotroph-like intermediary lobe peptide (CLIP) and β-MSH_5-22_ instead of ACTH [128,130,131]. As the ACTH precursors exhibit lower bioactivity, the ACTH to cortisol ratio is generally higher in patients with EAS than in those with Cushing’s disease and normal subjects. In addition to the ACTH precursors, elevated circulating levels of the hypothalamic neuropeptide agouti-related protein (AgRP) have been reported in EAS but not in Cushing’s disease. Therefore, AgRP has been suggested as a potential neuroendocrine tumor diagnostic marker [132]. In EAS, ACTH secretion is not suppressed through high-dose exogenous glucocorticoids, which is attributed to a defect in GR or GR-signaling [133]. This has been used during the diagnosis of EAS, and to differentiate it from Cushing’s disease. However, ectopic GR expression has been detected in several cases of EAS, which is a limitation of this test [134]. A dampened ACTH response to DDAVP has been observed in most EAS patients, whereas ACTH response is enhanced in Cushing’s disease, which is probably due to the ectopic AVPR1b or AVPR2 expression in these tumors [135,136]. However, in some EAS cases, AVPR1b has been reported to be expressed and respond to DDAVP [136,137,138,139]. Since the expression of CRH receptor has not been observed in most EAS cases, lack of ACTH secretion in response to CRH can be used as a reliable confirmatory test to diagnose EAS [140]. 

#### 3.3.2. Non-Neoplastic Hypercortisolemia

NNH, also known as pseudo-Cushing’s syndrome, is defined as an ACTH-dependent hypercortisolism state occurring in the absence of ACTH-secreting neoplasms. It is caused due to many clinical situations, such as excess alcohol intake, chronic kidney disease, depression, obesity, and poorly controlled Type 2 diabetes mellitus [141]. 

In patients with alcohol-induced hypercortisolemia, elevated midnight cortisol and high urinary free cortisol (UFC) levels with either elevated or normal ACTH levels have been reported. Low-dose DST failed to suppress the cortisol levels in these patients, making it difficult to clinically differentiate the illness from neoplastic hypercortisolemia. Alcohol can stimulate hypothalamic CRH and AVP secretion and impair hepatic clearance of cortisol [142,143,144]. In these patients, it is known that glucocorticoid resistance can be normalized through implementing alcohol abstinence [145].

Depression has been associated with impaired adequate termination of stress-induced HPA axis hyperactivity [146,147]. In these patients, both late-night cortisol and UFC levels are found to be elevated, and glucocorticoid sensitivity is reduced, which is similar to the clinical presentation of patients with Cushing’s disease. Impaired cortisol suppression following Dex administration was observed in 64% of patients with active depression [148]. Moreover, CRH administration failed to increase ACTH levels in these patients, probably due to low CRHR1 expression in corticotrophs caused by chronic CRH excess [149]. Although mechanisms underlying the pathology of this disease remain unclear, FKBP5 overexpression might be associated with HPA hyperactivity [150,151]. Moreover, in patients with depression and reduced glucocorticoid sensitivity, single nucleotide polymorphisms (SNPs) in the *FKBP5* gene leading to high FKBP51 expression have been identified [152].

Type 2 diabetes mellitus and obesity are associated with increased late-night salivary cortisol levels [153]. Aging, current DM, and high blood pressure have been associated with late-night salivary cortisol rather than a history of depression and current alcoholism [154]. Increased expression of 11-β-hydroxysteroid dehydrogenase 1 in adipose tissue might be a causal factor for the overproduction of cortisol in local tissue [155]. Cortisol hypersecretion is usually mild in these patients. However, it is unclear whether this cortisol elevation is a cause of metabolic syndrome rather than resulting from fat accumulation.

The DDAVP and/or combined DST/CRH tests have been used to distinguish patients with Cushing’s disease from those with non-tumoral hypercortisolism [138,156,157]. As normal corticotrophs express lower levels of AVPR1b, and intra-venous DDAVP injection does not stimulate ACTH secretion in patients who do not have ACTHomas, AVPR1b or AVPR2 has been shown to express abundantly in ACTHomas [105,106]. A combined DST/CRH test has also been applied to diagnose NNH. The rationale behind using this test is that the two conditions exhibit different sensitivity to low doses of dexamethasone and response to CRH [158]. Patients with NNH exhibit sensitivity to glucocorticoid-induced negative feedback, and therefore show a dampened response to CRH after Dex treatment. However, this test has not been useful in distinguishing alcohol-induced NNH [159]. It is therefore challenging to physiologically differentiate hyper-activity of the HPA axis from Cushing’s disease.

## 4. Conclusions

The molecular and genetic mechanisms underlying the pathophysiological regulation of ACTH secretion have been investigated in various clinical studies on patients with Cushing’s disease, EAS, and related conditions. Mainly, basic research has been performed using animal models and ACTH-secreting cell lines. Although there are several gaps in our understanding regarding these aspects, rapid progress due to recent technological advances, such as whole-exome sequencing, have enabled us to gain deeper insights into ACTH-related pathophysiology. However, the diagnostic methods and treatment of abnormal ACTH secretion are still limited, and thus, further investigations with a multifaceted approach are required to be performed in the future studies. 

## Figures and Tables

**Figure 1 ijms-21-09132-f001:**
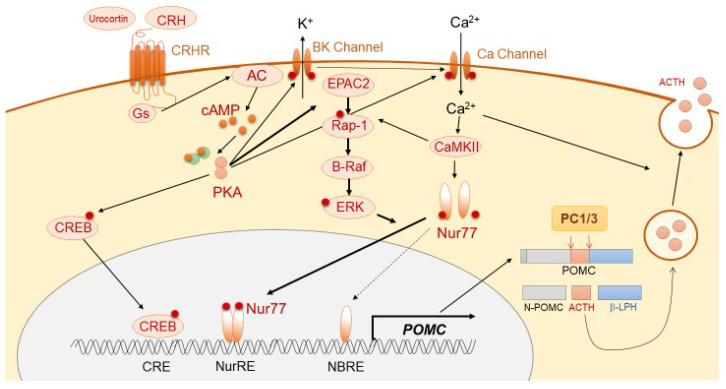
Schema of physiological ACTH synthesis and secretion.

**Figure 2 ijms-21-09132-f002:**
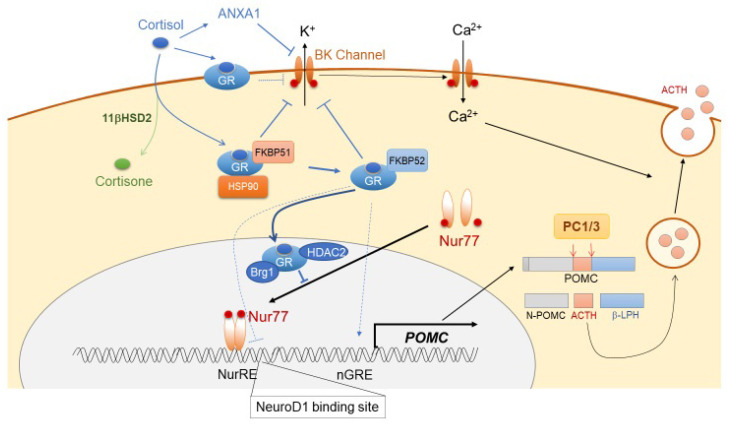
Schematic representation of glucocorticoid negative-feedback in corticotrophs.

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
