# Peer review of "The Mechanisms Underlying Autonomous Adrenocorticotropic Hormone Secretion in Cushing’s Disease"

_ijms, 2020, doi:10.3390/ijms21239132_

Round 1

Reviewer 1 Report

the work is well organized and comprehensive. I have no comments about the organization and the scientific value of the present review. 

however, I think that this paper is acceptable only after:

-a profound revision of the English language: I found numerous mistakes in grammar and syntax. I think that the entire paper has to be revised and rewritten in a more comprehensible English by a specialized revisor

-a different citation system. It is impossible that a list of references is itemized one by one in the text when indicated as a number between brackets. If the authors did not use a bibliographic reference manager, please, in the next version, use it (there are different free software options).

-even a review could show a material&methods section: this section should briefly explicit the search engine used to find bibliographic references (PUBMED, SCOPUS, WEB of SCIENCE, google...), the selected period of publications, and the MeSH terms used for the search (other people, reading this work and following this search description, should be able to find the same references). Given these premises, please add a material&method with such a description

Author Response

Dear Reviewer,

We are grateful to Reviewer 1 for the helpful comments on the original manuscript.  We have now addressed the critiques raised by you.

Comment 1.

-a profound revision of the English language: I found numerous mistakes in grammar and syntax. I think that the entire paper has to be revised and rewritten in a more comprehensible English by a specialized revisor

Response: Thank you for your kind review and the comment. We asked English editing to a special company namely Editage of this original manuscript and corrected it according to their edit. We have attached the proof of the English editing here.

Comment 2.

-a different citation system. It is impossible that a list of references is itemized one by one in the text when indicated as a number between brackets. If the authors did not use a bibliographic reference manager, please, in the next version, use it (there are different free software options).

Response: Thank you for pointing out. We have carefully checked one by one the text of the reference list and corrected according to the journal instruction.

Comment 3.

-even a review could show a material&methods section: this section should briefly explicit the search engine used to find bibliographic references (PUBMED, SCOPUS, WEB of SCIENCE, google...), the selected period of publications, and the MeSH terms used for the search (other people, reading this work and following this search description, should be able to find the same references). Given these premises, please add a material&method with such a description

Response: Thank you for your suggestion. According to the comment, we have added a material & methods section on Page 2 Line 58-63.

Reviewer 2 Report

In the present manuscript, Fukuoka et al. review basic principles as well as recently discovered mechanisms of ACTH secretion in Cushing's disease. The review is concise and well written despite some syntactically odd sentences that are scattered across the manuscript. The bibliography is essentially up-to-date which is not surprising given the recency of many of the discoveries around which the manuscript revolves. Overall, there is only little left to criticize and I can fully recommend publication after minor revisions of the following:

* page 3, lines 72ff.: The steps involved in POMC synthesis and cleaving of ACTH could be explained in some more detail.
* page 5, line 118: The citation contains #32 twice - is that intentional or is it a case of transposed digits?
* page 6, lines 141-143: This sentence needs language revision.
* page 8, line 190: The notion of up to 60% of CD cases to be due to USP8 mutations seems high and the true number more likely lies somewhere around one third of cases. Could you provide citations for these numbers?
* page 9, line 214: There is an "in" missing.
* page 10, lines 264-268: There are several concerns with the language (typos, syntax). Please revise carefully.
* page 11, line 283: "understood"
* page 11, line 289: "may be associated"
* page 12, line 303: language
* page 12, lines 309: Please use the abbreviation "AgRP" instead of "AGRP" (cf. line 308).
* page 12, lines 314-317: language
* page 13, line 355: ACTHomas (!)

Author Response

Dear Reviewer,

We are grateful to Reviewer 2 for the helpful comments on the original manuscript.  We have now addressed the critiques raised by you.

Comment 1.

The bibliography is essentially up-to-date which is not surprising given the recency of many of the discoveries around which the manuscript revolves.

Response: Thank you for your kind suggestion. We have added the list of bibliography on Page 9 Line 371 to Page 10 Line 373

Comment 2.

page 3, lines 72ff.: The steps involved in POMC synthesis and cleaving of ACTH could be explained in some more detail.

Response: Thank you for your comment. We totally agree with the reviewer and have added the following detail about POMC synthesis “The POMC transcriptional activity is thought to be regulated mainly via the orphan nuclear receptor, Nurr77, and also Tpit, Pitx, NeuroD1, signal transducer and activator of transcription 3 (Stat3), and ETS variant transcription factor 1 (Etv1) [4]. It is modulated by pituitary specific enhancer, which is located at -7 kb from the POMC initiation site [5], and also epigenetic modification, such as chromatin remodeling [6].” on Page 2 line 70 to 74 and cleaving of ACTH “POMC processing is also modulated by other enzymes, including Yapsin A, ACTH-converting enzyme (AACE), aminopeptidases B-like (AMB), and peptidylglycine a-amidating monooxygenase (PAM) [15].” on Line 88-90.

Comment 3.

page 5, line 118: The citation contains #32 twice - is that intentional or is it a case of transposed digits? 

Response: Thank you for your kind point out. We have corrected the twice citation.

Comment 4.

page 6, lines 141-143: This sentence needs language revision.

Response: Thank you for the helpful comment. We have corrected the language “ This is effectively negative feedback system of that operates within the HPA axis [45], and is involves both rapid nongenomic and delayed genomic mechanisms “ to “This is the predominant negative feedback system for the functioning of ACTH within the HPA axis [48], with both rapid non-genomic and delayed genomic mechanisms” as shown in page 4 line 146-148.

Comment 5.

page 8, line 190: The notion of up to 60% of CD cases to be due to USP8 mutations seems high and the true number more likely lies somewhere around one third of cases. Could you provide citations for these numbers?

Response: Thank you for the comment. According to reference [75]; Cell Res. 2015, 25, 306–317, doi:10.1038/cr.2015.20, the prevalence of USP8 mutation was reported as 62.04%. Therefore, we have addressed here 20 – 60% of ACTHoma cases here.

Comment 6.

page 9, line 214: There is an "in" missing.

Response: Thank you for the kind point out. We have added “in” on page 6 line 214.

Comment 7.

page 10, lines 264-268: There are several concerns with the language (typos, syntax). Please revise carefully.

Response: Thank you for the suggestion. We have corrected the sentence from “Recent high throughput screen newly developed a ACTH AlphaLISA assay identified a dual PI3K/HDAC inhibitor, CUDC-907 as a potential targeted therapy for ACTHomas. These though to be mediated via Nurr1 transcriptional activity in corticotroph adenomas, suggesting the important role of HDAC action in addition to PI3K in ACTHomas [105]” to “Newly developed high throughput “ACTH AlphaLISA assay” lead to the identification a dual PI3K/HDAC inhibitor, CUDC-907, as a potential targeted therapy for ACTHomas. It was thought to be mediated via Nurr1 transcriptional activity in corticotroph adenomas, suggesting an important role of HDAC action in addition to PI3K in ACTHomas [107]” as shown on page 7 line 260-263.

Comment 8.

page 11, line 283: "understood".

Response: Thank you for the kind point out. We have changed it.

Comment 9.

page 11, line 289: "may be associated"

Response: Thank you for the kind comment. We have revised it.

Comment 10.

page 12, line 303: language.

Response: Thank you for your suggestion. We agree with the comment. We have changed the sentence from “ In some carcinoids, ectopic PC2 expression can be explain the secretion of corticotroph-like intermediary lobe peptide (CLIP), and -MSH5-22 rather than ACTH [128][129][126][126][126].” to “For some carcinoids, ectopic PC2 expression can be explained as a reason for the secretion of corticotroph-like intermediary lobe peptide (CLIP) and -MSH5-22 instead of ACTH [128,130,131].” As shown on page 7 line 294-295.

Comment 11.

page 12, lines 309: Please use the abbreviation "AgRP" instead of "AGRP" (cf. line 308).

Response: Thank you for the point out. We have changed as reviewer suggestion.

Comment 12.

page 12, lines 314-317: language.

Response: Thank you for the kind comment. We agree with your suggestion. We have changed the sentence from “A blunted ACTH response to DDAVP has been observed in most EAS patients, whereas an increased ACTH response is observed in Cushing’s disease, probably due to a lack of AVPR1b or AVPR2 expression [133][134], while several cases of EAS has shown to express [134][135][136][137].” to “A dampened ACTH response to DDAVP has been observed in most EAS patients, whereas ACTH response is enhanced in Cushing’s disease, which is probably due to the ectopic AVPR1b or AVPR2 expression in these tumors [135,136]. However, in some EAS cases, AVPR1b has been reported to be expressed and respond to DDAVP [136–139].” as shown on page 7 line 304-308.

Comment 13.

page 13, line 355: ACTHomas (!).

Response: Thank you for the kind point out. We have corrected to ACTHomas.

Thank you for your valuable comments. We really appreciated your detailed suggestions.

Sincerely,

Hidenori Fukuoka

Round 2

Reviewer 1 Report

dear all, now the paper is ready to be published